# Perceptions, experiences, barriers, facilitators, learning outcomes, and modes of assessment of digital clinical placements for pre-registration physiotherapy students internationally: a systematic review protocol

**Justin McConnell** [1]*, **Alison Rushton** [1,2], **Tim Noblet** [1,2,3‡], **Verity Pacey** [3‡], **Jai Mistry** [1,2,3‡], **Jeremy Lai** [2‡], **Daphne Nguyen** [2‡], **Samantha Doralp** [1]

**1** Faculty of Health Sciences, University of Western Ontario, London, Ontario, Canada, **2** Physiotherapy Department, St George's University Hospitals Foundation Trust, London, England, United Kingdom, **3** Department of Health Sciences, Faculty of Medicine, Health and Human Sciences, Macquarie University, Sydney, New South Wales, Australia

☯ These authors contributed equally to this work.
‡ TN, VP and JM, JL, DN also contributed equally to this work.
* jmccon4@uwo.ca

## Abstract

### Introduction

The shift to digital clinical placements for physiotherapy education due to COVID-19 prompts a need for evaluation of current evidence. Existing studies highlight benefits of digital technology in clinical placements, but lack of a systematic review focused on pre-registration physiotherapy students is a key gap. This systematic review will address this gap by synthesizing the evidence for digital clinical placements for pre-registration physiotherapy students internationally.

### Methods and analysis

This systematic review is designed using the Preferred Reporting Items for Systematic Review and Meta-Analysis Protocols (PRISMA-P) statement and Cochrane Handbook – it is registered on PROSPERO (CRD42024571696). Search terms will be adapted to each database, including EMBASE, MEDLINE, PROSPERO, ERIC, and CINAHL. Key journals, forward citation tracking, references of included studies, and professional organization websites will also be searched. The search will include studies published since database inception to 31/05/24. There will be no limit to study design or language. Studies that report on perceptions, experiences, barriers, facilitators, learning outcomes, and modes of assessment of digital clinical placements for pre-registration physiotherapy students will be included. Meta-aggregation will be used to synthesize themes from findings which enables the generation of themes without the need to re-interpret data and the loss of study specific context.

**Data availability statement:** No datasets were generated or analysed during the current study. All relevant data from this study will be made available upon study completion.

**Funding:** The author(s) received no specific funding for this work.

**Competing interests:** The authors have declared that no competing interests exist.

## Ethics and dissemination

Ethics approval is not required. The results of this study will be written up for publication in relevant peer-reviewed scientific journals and contribute to a developing area of research. Results will also be presented at national or international conferences, events for the physiotherapy profession, or education events.

## Introduction

To become a physiotherapist, pre-registration physiotherapy students need to gain practical experience in clinical settings working with patients under the supervision of experienced professionals, commonly termed clinical educators, clinical instructors, or preceptors. Traditionally, this practical experience is gained through in-person clinical placements which enables hands-on training and exposure to real-world situations. However, with the COVID-19 pandemic and advancements in digital technology, the traditional approach to clinical placements was disrupted and digital clinical placements have emerged as a viable and appropriate alternative [1–4]. As digital clinical placements increasingly become more prevalent in the education of pre-registration physiotherapy students, it is necessary for physiotherapy educational programs to develop physiotherapists with understanding, skills, and knowledge in digital technologies across practice settings as called for by World Physiotherapy [5]. This review will explore the current evidence on digital clinical placements for pre-registration physiotherapy students. But first, it is necessary to operationalize the term "digital clinical placements" to ensure clarity in this review because there are various terms and concepts used internationally and throughout the literature.

### Defining digital clinical placements

Several terms are used in literature to describe the concept of digital alternatives as a different way to offer clinical placements or to supplement them. For example, Sam et al. [6] and Houghton et al. [7] describe "digital clinical placements" as a scenario where in-person placements for medical students are offered with asynchronous or synchronous clinical education materials and resources (e.g., problem-based learning, online case studies, educational videos, Q&A's, live webinars, and self-directed tutorials) delivered through a digital format. Quail et al. [8] describe the supplementation of in-person placements with simulation through Virtual Learning Environments (VLEs) and Standardised Patients (SPs) as "virtual clinical placements." VLEs are computer generated applications created to achieve a specific learning outcome and SPs are human actors trained to respond in specific ways to students participating in a simulation (e.g., following a script or expressing specific symptoms) [8]. Robinson et al. [9] describes Simulated Learning Environments (SLEs) as both "virtual and digital patient placement" – SLEs use SPs and problem-based learning. Twogood et al. [1] and Vrzic et al. [3] also use the term "virtual clinical placement" to describe physiotherapy students participating in live patient consultations through digital communication (i.e., telehealth). Heneghan et al. [2] use the term "telehealth e-mentoring" to describes post-graduate physiotherapists using telehealth alongside their mentors to deliver consultation and rehabilitation. There is incongruent use of terms in the literature, as such, operationalizing a term will distinguish these varied descriptions of similar but distinct concepts to facilitate straightforward understanding and clarity for this review. A shared understanding enhances the coherence of academic and professional discourse and aids in the development of shared knowledge bases. As Podsakoff et al. [10] stated, "… concepts serve as the fundamental building blocks of theory, allowing us

to organize complex phenomena with a common language that, when done well, facilitates communication between researchers. (p. 168)" For this paper, "digital clinical placements" will be described as healthcare delivered a/synchronously by pre-registered healthcare professionals (HCPs) under the supervision of registered HCPs to human patients entirely or partially via information and communication technology. This definition allows for clarity in defining inclusion criteria and the search strategy of this review.

## The state of evidence

A preliminary search using Google Scholar, Cochrane Library, and Physiotherapy Evidence Database (PEDro) identified two reviews on digital clinical placements for pre-registration physiotherapy students with limited physiotherapy perspective and methodological rigour. First, Bridgman et al. [11] conducted a synthesis of published literature on the educational perspectives of allied health students on clinical placements using telehealth. Their findings suggest that telehealth placements offer financial, accessibility, and educational support benefits, however, considerable work is needed to address challenges in the overall experience [11]. Second, Romli et al. [12] aimed to determine if healthcare students acquired sufficient competence through alternative clinical placements such as telehealth, simulation, and case-based learning. Their systematic review suggests alternative clinical placements are either more effective or just as effective as traditional placements [12]. Both Bridgman et al.'s and Romli et al.'s reviews are limited in their capture of the physiotherapist perspective. A single paper with physiotherapy perspective is included in each review, 1 of 3 in the Bridgman article and 1 of 24 in the Romli article, this raises concerns for the generalizability of their findings towards physiotherapy.

Additionally, findings from Bridgman et al.'s and Romli et al.'s reviews should be accepted cautiously because of appraisal ratings of *critically low* and *low* [13], respectively. Both reviews used search strategies that included data bases only and raise concerns about the rigour of their methods. Further methods of searching such as forward citation tracking, searching reference lists, or manually searching specific journals and professional organizations websites would improve the design by increasing the likelihood of capturing all relevant data [14]. Romli et al.'s work remains important as the attainment and assessment of competency is an important aspect of education. Clinical competency is a precursory requirement of entry-to-practice for pre-registration physiotherapy students [15]. Therefore, evidence surrounding attainment and assessment of competency is valuable for informing physiotherapy educational programs. Both reviews underscore the need for a more rigorous review that is specific to physiotherapy to produce relevant and applicable findings. Therefore, a high-quality systematic review is required to synthesise the current evidence in the evolving field of digital clinical placements for pre-registration physiotherapy students.

Within this evolving field several primary studies [1,3,16–19] highlight the opportunities and challenges associated with digital clinical placements for pre-registration physiotherapy students. Students reported initial skepticism about the efficacy of telehealth in meeting learning objectives, particularly for manual therapies and hands-on skills [16]. Students also reported positive perceptions regarding telehealth. For example, Ross et al. [16] and Davies et al. [19] highlighted that students found telehealth beneficial for addressing accessibility and for follow-up consultations. A majority of students also advocated for a blended model combining in-person and telehealth [19]. While these studies provide valuable insight into the perceptions of physiotherapy students, generalizability is limited because the findings of each study focus on perceptions of students from only one university each – University of Queensland in Brisbane, Queensland, Australia and Macquarie University in Sydney, New South Wales, Australia.

As for clinical educators, some in Brisbane, Queensland, Australia believe digital clinical placements provided a unique opportunity for students to develop clinical reasoning, communication, and planning skills [17]. However, these clinical educators also believed that hands-on skills were less developed, and client outcomes were varied depending on client engagement and technological proficiency [17]. In regards to the clinical educator context of digital clinical placements, the same clinical educators from Brisbane reported challenges in efficient supervision, technical issues, and adapting educational models for digital [17]. These clinical educators also noted that telehealth allowed flexibility in placements, including remote supervision, though it required careful client selection and session preparation [17]. Another challenge noted by a clinical educator at a pain management clinic in Kingston, Ontario, Canada was navigating supervision without clear guidelines for digital healthcare [3]. One group in Australia and New Zealand aimed to address this same concern. This group created a guide on how to interpret the Assessment of Physiotherapy Practice (APP), the assessment tool used on clinical placements for physiotherapy students, to better evaluate competencies in the telehealth context [18]. A strength of this expert group was that it included the perspectives of several clinical educator leaders representing several education institutions and their respective clinical educators. This guide can be used to inform clinical educators assessing student skills within a digital clinical placement [18], however the applicability outside of the Australia and New Zealand context should be carefully assessed.

For the institutional context, there appears to be agreement on opportunities and challenges with the added perspective of sustainability and scalability of digital clinical placements. Twogood et al. [1] reported digital clinical placements had strengths in accessibility for students, particularly those with health risks and living abroad, and in opportunities to develop digital care skills essential to the evolving healthcare landscape. Twogood et al. [1] also reported challenges in technical issues and limited opportunities for hands-on clinical practice. Additionally, Twogood et al. [1] reported administrative challenges in scheduling and communication. Despite these challenges, digital clinical placements increased clinical placement capacity by over 400% compared to pre-COVID periods [1]. Twogood et al. [1] note the opportunity for scalability, but recognize the sustainability of this model may require adaptation for different contexts. Digital clinical placements are a vital innovation in physiotherapy education. Despite challenges, they offer a viable opportunity to expand placement options and equip students with essential digital healthcare skills. This systematic review will synthesize current evidence to provide insights that guide future integration of digital clinical placements into physiotherapy education.

## Aim

To investigate the perceptions, experiences, barriers, facilitators, learning outcomes, and modes of assessment of digital clinical placements for pre-registration physiotherapy students internationally

## Objectives

1. To identify and synthesize perceptions and experiences associated with digital clinical placements for pre-registration physiotherapy students from the perspective of physiotherapy educational programs, pre-registration physiotherapy students, and physiotherapy clinical educators.

2. To identify and synthesize barriers and facilitators associated with digital clinical placements for pre-registration physiotherapy students.

3. To identify and synthesize learning outcomes and modes of assessment currently being used for digital clinical placements for pre-registration physiotherapy students.

## Methods

### Positionality

Researcher ontological, epistemological, and methodological stances shape the design, conduct, and interpretation of studies. Our ontological perspective lies between realism and relativism, acknowledging that while an external reality may exist, our understanding of it is inevitably influenced by individual experiences and social contexts. This position allows us to recognize the shared realities within digital clinical placements for pre-registration physiotherapy students while also appreciating the diversity of individual and contextual experiences. Epistemologically, our stance aligns with interpretivism, with elements of positivism and constructivism. This means we will prioritize understanding the subjective meanings and experiences of individuals engaged in digital clinical placements, while also acknowledging the value of systematically synthesizing findings to identify patterns and broader truths. We embrace a constructivist approach in appreciating that knowledge can be co-constructed between researchers and participants, yet we remain open to positivist ideals when aiming for rigor and reproducibility in the synthesis of evidence. Methodologically, we have chosen a qualitative systematic review with meta-aggregation for data synthesis. This approach aligns with our philosophical stance by enabling us to systematically identify, appraise, and synthesize qualitative evidence. Meta-aggregation offers a structured way to collate findings without losing the original studies interpretation, ensuring the synthesis reflects both the shared and unique aspects of the phenomenon under study. We are aware that our professional background and experiences influence how we approach and interpret the data. As professors, PhD students, and physiotherapists researching digital clinical placements for pre-registration physiotherapy students, we bring insights from both academic and practical sides of the field. We strive to remain reflexive throughout the research process, continuously interrogating our biases and assumptions to ensure they do not unduly influence the synthesis of the findings. Our aim is to conduct research that is rigorous, meaningful, and grounded in the lived realities of those it seeks to represent.

### Design

This systematic review is designed using a combination of the Preferred Reporting Items for Systematic Review and Meta-Analysis Protocols (PRISMA-P) 2015 statement [20] and Cochrane Handbook [14]. It is registered with the International Prospective Register of Systematic Reviews (PROSPERO CRD42024571696).

### Eligibility criteria

Eligibility criteria were created using a modified PICOS framework to ensure sensitivity during the search process [21]. Inclusion criteria (Table 1) were created to ensure the search is as broad as possible. From the framework, Comparator was not included as it is not applicable to the research aim and objectives.

### Information sources

Information sources will be searched from their inception to the 22nd of January 2025. The sources of information that will be searched include:

**Table 1. Inclusion criteria.**

| Inclusion criteria |
| --- |
| *Population* |
| Students |
| • Students currently or previously enrolled in a pre-registration physiotherapy program (BScPT, MPT, DPT). |
| Clinical Instructors/Educators |
| • Clinical instructors that have experience supervising pre-registration physiotherapy students in a clinical setting. |
| Educational Institutions |
| • Faculty/instructors or administrative staff who contribute to the clinical education component of a pre-registration physiotherapy program (BScPT, MPT, DPT). |
| *Intervention* |
| Digital clinical placements defined as clinical placements which: |
| • Entirely or partially using digital technology and methods (e.g., telehealth, telerehabilitation, telecare, teleconsult, telemedicine, and remote nonclinical service). |
| • Synchronous or asynchronous |
| *Outcomes* |
| Outcome measures including: |
| • Perceptions and experiences |
| • Barriers and facilitators |
| • Learning outcomes |
| • Modes of assessment |
| *Study* |
| Study designs including: |
| • Primary research: Any design that is qualitative or mixed methods (reporting qualitative data) |
| Publication language |
| • Any language |

**Databases.** EMBASE, MEDLINE, PROSPERO, Educational Resource Information Centre (ERIC), and Cumulative Index to Nursing and Allied Health (CINAHL) databases will be included in the search strategy.

**Specific journals.** Key journals will be manually searched and include Physiotherapy, Physiotherapy Canada, Physical Therapy & Rehabilitation, Journal of Physiotherapy, and Physiotherapy Theory and Practice.

**Professional organizations.** Relevant articles available on professional organizations' (e.g., physiotherapy associations and special interests' groups) websites will be searched manually. This will be achieved by entering relevant phrases adapted from the search strategy into Google and then manually searching the first 100 results [22].

**Forward citation tracking.** Forward citation tracking will be used as a method to identify newer publications that have cited a particular article or other scholarly work. Included studies will be entered in Google Scholar to conduct forward citation tracking. While similar to Web of Science and Scopus, Google Scholar will be used as it can capture unique citations including non-English sources [23].

**Searching reference lists.** The reference lists of included studies will be searched [14]

## Search strategy

The search strategy was created in collaboration with a research librarian at University of Western Ontario Libraries. Three concepts were chosen to ensure the search strategy is as broad as possible. A complete search strategy is demonstrated in Table 2. Database specific subject headings (i.e., MeSH terms used in the MEDLINE OVID platform) will be combined with concepts where possible. The MEDLINE OVID platform search strategy is presented in S1 Appendix. The search strategy will be adapted for each information source.

**Table 2. Search keywords for search strategy.**

| Concept | Keywords/Synonyms |
|---------|-------------------|
| "Digital placement" | "digit*" OR "cyber" OR "distance*" OR ("e?care" or "e?consultation*" or "e?health" or "e?healthcare" or "e?medicine" or "e?monitor*" or "e?service*" or "e?visit*") OR ("electronic" and ("care" or "consultation*" or "health" or "healthcare" or "medicine" or "monitor*" or "service*" or visit*")) OR "mobile*" OR "online*" OR "remote*" OR "tele*" OR "videoconference*" OR "virtual*" OR "web-based*" |
| Physiotherapist | "physi* therap*" OR "physiotherap*" |
| Pre-registration | "pre-registration*" OR "apprentice" OR "candidate*" OR "entry-level*" OR "intern*" OR "learner*" OR "pre-licen*" OR "pre-professional*" OR "probationer*" OR "resident*" OR "school-based" OR "student*" OR "trainee*" |

## Study records

**Data management.** Results from all searches will be imported to, stored, and managed on Covidence [24]. The reference, abstract, and full text file will be stored on Covidence for screening. Duplicates will be removed prior to commencement of the selection process.

**Selection process.** Two researchers will independently and simultaneously review titles and abstracts of the identified articles using Covidence's structured workflow [25]. The predetermined eligibility criteria (Table 1) will be used to separate the articles into one of three categories: eligible, not eligible, and potentially eligible. Articles that are obviously not related to the objective based on their title and abstract will be excluded. If an article is unable to be excluded based on title and abstract, then it will be categorised as potentially eligible. After screening abstracts and titles, full-text copies of potentially relevant articles will be obtained, and eligibility ascertained using the same process. If there is disagreement between researchers regarding selection of studies, then a third researcher will be consulted. The selection process will be presented visually per PRISMA 2020 Guidelines [20]. The Kappa coefficient will be used to evaluate the inter-rater reliability between reviewers [26].

**Data collection process.** Two researchers will independently extract qualitative data using a study-specific data extraction grid within Microsoft Excel. If data is unavailable in the full-text of included studies, then the original authors will be contacted to ask for a copy of the original study's data. In the event of disagreement or lack of consensus between two researchers, then a third researcher will be consulted to make the decision for inclusion.

## Data items

The data extraction grid will include key data items: citation details (title of study, authors, year of publication, and country), study design, population, purpose/phenomenon of interest, aims and objectives of the study, philosophical positioning, methodology/methods, and key findings. Key findings will be identified by the themes presented in the results of included studies. For example, this may include the subsection titles in the results section or may include the authors explicitly stating the themes of their study. These themes will then be used for meta-aggregation, a data synthesis method that does not use interpretive analysis of extracted data from included studies, instead it enables a summary of the original authors' findings across included studies to determine common or competing themes [27]. The 'Data synthesis' section explains further how meta-aggregation avoids re-interpretation.

## Quality assessment

Two researchers will independently use the Quality Assessment with Diverse Studies (QuADS) tool [28] to assess methodological and reporting quality of included studies. The

QuADS is a tool for the assessment of quantitative, qualitative, and mixed-methods studies that has an inter-rater reliability with a Kappa coefficient of k = 0.66 [28]. The QuADS and will provide a structured approach to quality assessment that is user friendly for the authors of this study. The QuADS scores will be reported in a table. If there is a disagreement between the two researchers, then a third researcher will be consulted. The quality assessment ensures transparency and confidence in the quality of the included studies' findings.

## Data synthesis

Meta-aggregation will be used to synthesize extracted data. Meta-aggregation is a three-step synthesis model that avoids re-interpretation of data from included studies and aims to accurately and reliably present the findings of the authors from the included studies [27]. Avoiding re-interpretation ensures the results remain with the specific contextual factors that informed the original authors' interpretation of the data. The first stage is a detailed extraction of findings from the results section of included studies and will be presented in a table. The second stage is categorization of the findings by identifying a common theme between at least two findings per category [27]. Categories will be based on similarity of concepts between the findings and developed using a consensus process of at least two researchers. The third stage involves developing a synthesized finding of at least two categories [27] that will aim to capture the collective meaning of the categories. Two researchers will develop the synthesized finding through a consensus process with a third researcher having final input. The findings will be illustrated as shown in the example in Fig 1.

## Confidence in cumulative evidence

GRADE-CERQual (Confidence in the Evidence from Reviews of Qualitative research) will be used to assess quality and strength of evidence found in this study. GRADE-CERQual is a structured method for determining the confidence in evidence used for qualitative data synthesis and usefulness of the findings [29]. GRADE-CERQual assesses four components in each synthesis finding: methodological limitations, coherence, adequacy of data, and relevance [29]. Methodological limitations will be assessed using the QuADS tool as described in the *Quality assessment* section [30]. Coherence, adequacy of data, and relevance will be assessed with guidance from key papers outlining how to apply GRADE-CERQual to qualitative evidence synthesis [29–34]. Coherence is how clear and compelling the fit is between primary research data and a synthesized review finding. Coherence will be assessed using three steps: 1) collect and consider the necessary information related to coherence (i.e., clear patterns across primary research data that supports the review finding), 2) assess the body of data that contributes to each review finding and decide whether there are concerns about coherence, and 3) make a judgement about the seriousness of the concerns and justify this judgement [31]. Adequacy of data is a measurement of data richness and data quantity supporting a review finding. Adequacy of data will be assessed using three steps; 1) collect and consider the necessary information related to adequacy (i.e., the amount and depth of data that relates to the objectives of this review), 2) assess the body of data that contributes to each review finding and decide whether there are concerns about adequacy, and 3) make a judgement about the seriousness of the concerns and justify this judgment [32].

Relevance will be assessed in five steps; 1) clarify the review question and context, 2) decide on the appropriateness and implication of the sampling strategy, 3) gather information about relevance in the included studies, 4) assess the body of data that contributes to each review finding and decide whether there are concerns about relevance, 5) make a judgement about the seriousness of the concerns and justify this judgement [34]. Table 3 is an example of the table that will be

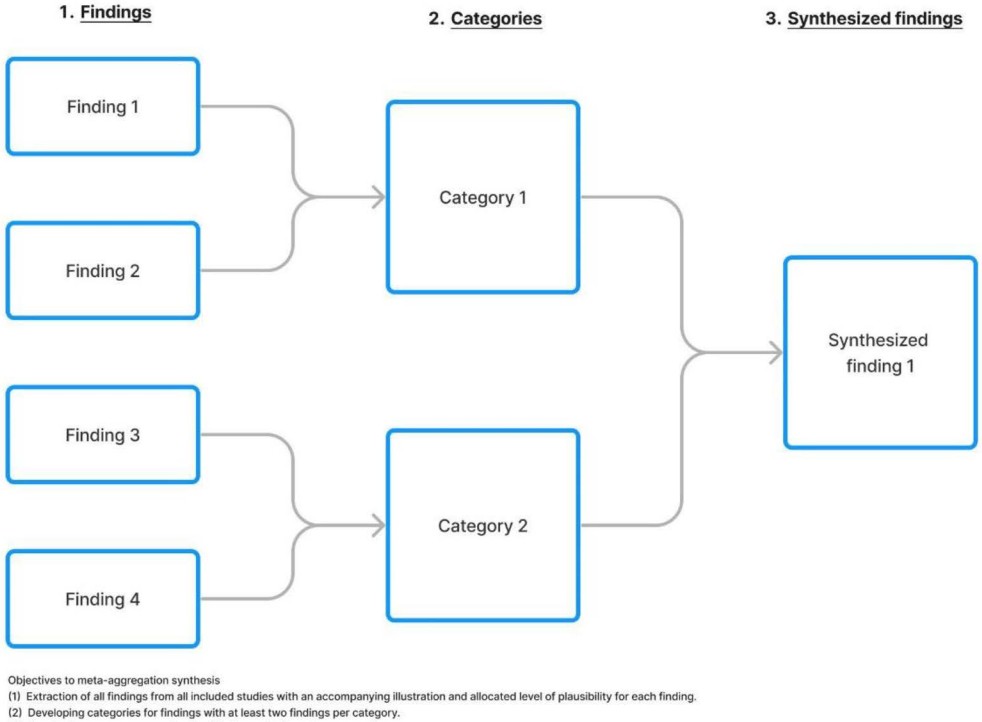

Objectives to meta-aggregation synthesis
(1)  Extraction of all findings from all included studies with an accompanying illustration and allocated level of plausibility for each finding.
(2)  Developing categories for findings with at least two findings per category.
(3)  Developing one or more synthesized findings of at least two categories.

**Fig 1. Example illustration of meta-aggregation process.**

created to facilitate the review findings for each objective. The assessments will be assigned one of four rankings: *no or very minor concerns*, *minor concerns*, *moderate concerns*, or *serious concerns* to each finding of this study [33]. A *no or very minor concerns* ranking would indicate that the review finding is likely be an appropriate representation of the phenomenon of interest and a *serious concerns* ranking would indicate that the review finding is very unlikely to be an appropriate representation of the phenomenon of interest [33]. An overall confidence in the evidence rating will be assigned as *high, moderate, low,* or *very low*. A *high* ranking would indicate that more research is unlikely to affect the confidence of the review findings and a *very low* ranking would indicate there are concerns that are very likely to reduce confidence of the review findings [33].

## Publication and dissemination of results

The results of this study will be published in an international peer-reviewed scientific journal and contribute to a developing area of research. Results will also be presented at national or international peer-reviewed conferences, events for the physiotherapy profession, and education events.

## Resources

The author(s) received no specific funding for this work.

## Implications of this study

The authors of this review anticipate that the findings will provide insight into the perceptions and experiences of pre-registration physiotherapy students, clinical instructors, and

**Table 3. Confidence in cumulative evidence.**

**Objective: Objective 1 written here**

| Review Finding | Studies Contributing to the Review Finding | Methodological limitations | Coherence | Adequacy of Data | Relevance of the Data | CERQual Assessment of Confidence in the Evidence | Explanation of CERQual Assessment |
|---|---|---|---|---|---|---|---|
| 1. Review finding | 1, 2, 3,4, … | No or very minor/minor/moderate/serious methodological concerns. Comments will be informed by the quality assessment conducted using the QuADS tool. | No or very minor/minor/moderate/serious coherence concerns. Step 1: Access the data contributing to the review finding. This will be found in the data extraction table. Step 2: Identify if the underlying data is contradictory, ambiguous/plausible alternative, or incomplete/insufficient. Step 3: Make a judgement about the seriousness of concerns and justify this judgement | No or very minor/minor/moderate/serious adequacy of data concerns. Step 1: Access the data contributing to the review finding. This will be found in the data extraction table. Step 2: Assess the richness of the data relative to being descriptive or explanatory. Assess data quantity (i.e., number of participants and number of studies). Step 3: Make a judgement about the seriousness of concerns and justify this judgement | No or very minor/minor/moderate/serious relevance of data concerns. Step 1: Access the data contributing to the review finding. This will be found in the data extraction table. This data will be assessed against the aim & objectives as well as the inclusion criteria of this review Step 2: Assess the appropriateness and implications of the included studies against the review question. Step 3: Gather information about relevance in the included studies. This will be achieved through data extraction and the data extraction table. Step 4: Assess the data for its relevant contribution to the review finding and decide if there are concerns about this relevance. Three threats to relevance should be considered, data has indirect relevance, partial relevance, or unclear relevance. Step 5: Make a judgement about the seriousness of concerns and justify this judgement | High/moderate/low/very low confidence | A summarization and explanation for the confidence rating given the previous column. |

A summarized version of Table 3 will also be created (Table 4) to highlight the overall assessment of confidence and whether the finding of the synthesis are a reasonable representation of the phenomenon of interest [29].

**Table 4. Summary of confidence in cumulative evidence.**

**Objective: Objective 1 written here**

| Review Finding | Studies Contributing to the Review Finding | CERQual Assessment of Confidence in the Evidence | Explanation of CERQual Assessment |
|---|---|---|---|
| 1. Review finding | 1, 2, 3,4, … | High/moderate/low/very low confidence | A summarization and explanation for the confidence rating given the previous column. |

physiotherapy educational faculty involved in digital clinical placements. This review will provide insight to the barriers and facilitators to the success of digital clinical placements. This review will provide insights regarding current learning outcomes and modes of assessment being used for digital clinical placements for pre-registration physiotherapy students. The authors expect the findings will be specific and relevant to educational institutions and research groups interested in digital clinical placements for pre-registration physiotherapy students, as well as clinicians and clinical sites supporting students on clinical placements. This review will identify gaps in the evidence and suggest future opportunities for research. The findings may underpin curriculum decision-making for educational programs conducting digital clinical placements.

## Supporting information

**S1 File. PRISMA-P 2015 checklist.**
(PDF)

**S1 Appendix. OVID medline search strategies.**
(PDF)

## Author contributions

**Conceptualization:** Justin McConnell, Alison Rushton, Samantha Doralp.

**Data curation:** Justin McConnell.

**Formal analysis:** Justin McConnell, Jeremy Lai, Daphne Nguyen.

**Investigation:** Justin McConnell, Jeremy Lai, Daphne Nguyen.

**Methodology:** Justin McConnell, Alison Rushton, Samantha Doralp.

**Project administration:** Justin McConnell.

**Resources:** Justin McConnell.

**Supervision:** Alison Rushton, Tim Noblet, Samantha Doralp.

**Validation:** Justin McConnell.

**Visualization:** Justin McConnell, Samantha Doralp.

**Writing – original draft:** Alison Rushton, Samantha Doralp.

**Writing – review & editing:** Alison Rushton, Tim Noblet, Verity Pacey, Jai Mistry, Jeremy Lai, Daphne Nguyen.

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
