## [Decision Letter · Decision Letter 0]

15 Nov 2024

PONE-D-24-34705Perceptions, experiences, barriers, facilitators, learning outcomes, and modes of assessment of digital clinical placements for pre-registration physiotherapy students internationally: a systematic review protocolPLOS ONE

Dear Dr. McConnell,

Thank you for submitting your manuscript to PLOS ONE. After careful consideration, we feel that it has merit but does not fully meet PLOS ONE’s publication criteria as it currently stands. Therefore, we invite you to submit a revised version of the manuscript that addresses the points raised during the review process.

Please submit your revised manuscript by Dec 30 2024 11:59PM. If you will need more time than this to complete your revisions, please reply to this message or contact the journal office at plosone@plos.org . Please include the following items when submitting your revised manuscript:

We look forward to receiving your revised manuscript.

Kind regards,

Catherine Doody, PhD

Academic Editor

PLOS ONE

**Journal Requirements:**

Reviewers' comments:

Reviewer's Responses to Questions

**Comments to the Author**

1. Does the manuscript provide a valid rationale for the proposed study, with clearly identified and justified research questions?

Reviewer #1: Yes

2. Is the protocol technically sound and planned in a manner that will lead to a meaningful outcome and allow testing the stated hypotheses?

Reviewer #1: Yes

3. Is the methodology feasible and described in sufficient detail to allow the work to be replicable?

Reviewer #1: Yes

4. Have the authors described where all data underlying the findings will be made available when the study is complete?

Reviewer #1: Yes

5. Is the manuscript presented in an intelligible fashion and written in standard English?

Reviewer #1: Yes

6. Review Comments to the Author

You may also provide optional suggestions and comments to authors that they might find helpful in planning their study.

**Reviewer #1&2 :**  Thanks for the invitation to review this manuscript. For context, this review has been completed asynchronously by two academics, and then discussed collectively to evaluate the quality of our review. We’ve labelled our comments below accordingly for your context. The overarching topic is interesting and relevant for the physiotherapy profession and broadly digital healthcare, and the methodological approach is excellent. Overall, we believe the outcomes of this review will offer important contributions to the study and practice of digital clinical placements for the physiotherapy profession. Nevertheless, below we report several comments for your consideration, which are essentially ‘minor’ in nature. We genuinely consider the review process an academic debate, so encourage you to think critically about our feedback and offer theoretically and/or empirically justified responses where you disagree with our recommendations/comments.

1 R1: I appreciated reading an entire section devoted to the definition of digital clinical placements. As you acknowledge, the diversity of definitions within existing literature indicates the need for clarity, at least how it pertains to the current work. However, I wonder if you could tighten your current approach, so that a single statement captures the essence of this concept rather than the definition being spread across three individual statements. For example, perhaps something like “healthcare delivered a/synchronously by trainees under the supervision of registered professionals to human patients entirely or partially via information and communication technology” might work well. FYI, this paper by Podsakoff and colleagues (https://journals.sagepub.com/doi/full/10.1177/1094428115624965) is my ‘go to’ reference when it comes to thinking about high-quality concept definitions.

2. R2: In the section titled ‘the state of the evidence’, the evidence provided comes from two reviews on this topic, each of which references a single article pertaining to physiotherapy student placements. Despite this being a relatively under-explored topic of research in the physiotherapy profession, there are other journal articles (not captured by the aforementioned reviews) that could be included in this section to provide a more comprehensive overview of the topic of interest. Here are just a few examples, within physiotherapy, that could be considered for inclusion:

• https://doi.org/10.5195/ijt.2022.6464

• https://doi.org/10.53300/001c.32992

• https://doi.org/10.53300/001c.24960

• https://doi.org/10.11157/fohpe.v23i3.595

• https://doi.org/10.1002/msc.1723

• https://doi.org/10.1002/hsr2.2067

R1: I have a couple of reflections regarding the section titled ‘the state of the evidence’:

• Line 114: a single paper among how many? This context is important to help readers appreciate the essence of your reflections on their work.

• I believe you could sharpen your critical evaluation of their methodology to strengthen the gap you address here. For example, authors of this prior work may have set out to scope perspectives broadly across allied health rather than in-depth for any specific discipline. If so, your critical evaluation could be considered a strength to some extent rather than a weakness, or at least an artifact of their guiding research questions (I suspect the search strategy to capture everything across all health sciences would be complex and capture a lot of irrelevant work). Expanding the points you mention also seems important. For example, what is the value of the additional search strategies you mention (e.g., increase the likelihood that we capture most if not all relevant data)?

3. R2: In objective 1 you state ‘digital/online clinical placements’ whereas objectives 2 and 3 simply state ‘digital clinical placements’. In light of your stated definition of ‘digital clinical placements’ (lines 91-92), it would be appropriate to delete the word ‘online’ in objective 1 for consistency.

4. R1: General writing comment: people are ‘who’ or ‘whom’ rather than ‘that’ (e.g., Table 1 “Clinical educators that have experience…”).

5. R2: Minor typographic errors:

• In Table 1 Inclusion criteria (page 14): “… using …” should read “… uses …”

• Line 294: “asses’ should read “assess”

• Table numbers appear incorrect i.e., there are two (2) Table 1’s referenced within the body of the paper (page 14 and page 26)

• In Table 1 Confidence in Cumulative Evidence (last column on page 26) and Table 2 Summary of Confidence in Cumulative Evidence (last column on page 27): “… rating give in the …” should read “… rating given in the …”

6. R1 and R2: The systematic review protocol is excellent – well done. We have a few reflections for your consideration:

• Regarding data items, you might consider adding philosophical positioning to the list, as you’ll likely access mostly qualitative studies in which this information is central to assessments of rigor. Relatedly, you also might like to add your philosophical positioning for this work so that readers can appreciate key concepts like ontology and epistemology as it relates to your efforts.

• Since you are targeting assessment modes, learning outcomes, beliefs, etc…, it is possible these aspects could be overlooked (e.g. if captured via standardized self-reports) if only accessing qualitative and mixed methods designs. As such, you could consider expanding the inclusion criteria to also include Quantitative studies. Alternatively, you may need to provide a rationale for why you are excluding quantitative only studies and edit the research objectives accordingly.

• Lines 237-240: This statement could be strengthened so that readers can appreciate exactly what you plan to do here. I’m unsure that one can collate and summarise ‘themes’ among a body of work with no interpretation on the part of analysts. You might not be interpreting the original work per se but nevertheless are ‘filtering’ that information in some way (i.e., you are an active driver rather than passive passenger). Relatedly, what exactly do you mean by a theme here?

7. R1: I preface this comment with my general impression that the QuADS and GRADE-CERQual appear reasonable enough for your purposes. Nevertheless, I also would like to add that quality appraisals and confidence in evidence for qualitative research are notoriously complex and, therefore, ‘one size fits all’ approaches are typically considered inadequate by qualitative experts (for which I am not!). As one example, Smith and McGannon eloquently explained the complexities of rigour for qualitative research in this paper (https://www.tandfonline.com/doi/full/10.1080/1750984X.2017.1317357). Brett Smith and others have written extensively on such issues (e.g., https://doi.org/10.1016/j.psychsport.2009.02.006, https://www.tandfonline.com/doi/full/10.1080/1612197X.2019.1637363).

8. R2: Regarding data synthesis, I fully appreciate the desire to focus your meta-aggregation on commonalities among existing work. Nevertheless, as a thought-provoking comment, you also might like to consider ‘exceptions’ within the existing body of work and how they might shed light on your research questions. Phoenix and Orr wrote an excellent piece that might provide inspiration for you (https://www.tandfonline.com/doi/full/10.1080/2159676X.2017.1282539).

7. PLOS authors have the option to publish the peer review history of their article (what does this mean? ). If published, this will include your full peer review and any attached files.

**Do you want your identity to be public for this peer review?** For information about this choice, including consent withdrawal, please see our Privacy Policy .

Reviewer #1: No

---

## [Author Response · Author response to Decision Letter 0]

30 Dec 2024

Hello PLOS One Team, we received an email regarding revision to our manuscript. We appreciate the editor and reviewer taking the time to review our manuscript and have revised it as suggested. Please find in our revision submission the following items:

• A rebuttal letter that responds to each point raised by the academic editor and reviewer(s). We have uploaded this letter as a separate file labeled 'Response to Reviewers'.

• A marked-up copy of our manuscript that highlights changes made to the original version. We have uploaded this as a separate file labeled 'Revised Manuscript with Track Changes'.

• An unmarked version of our revised paper without tracked changes. We have uploaded this as a separate file labeled 'Manuscript'.

Thank you again.

---

## [Decision Letter · Decision Letter 1]

26 Jan 2025

Perceptions, experiences, barriers, facilitators, learning outcomes, and modes of assessment of digital clinical placements for pre-registration physiotherapy students internationally: a systematic review protocol

PONE-D-24-34705R1

Dear Dr. McConnell

We’re pleased to inform you that your manuscript has been judged scientifically suitable for publication and will be formally accepted for publication once it meets all outstanding technical requirements.

Kind regards,

Catherine Doody, PhD

Academic Editor

PLOS ONE

Additional Editor Comments (optional):

Thank you for your correspondence in relation to altering the date of your proposed systematic review search.

I have noted this and have sent it on the section Editor as an important amendment;

Line 250 replace ‘to the 31st of May 2024’ with “to the 22nd January 2025”.

Reviewers' comments:

Reviewer's Responses to Questions

**Comments to the Author**

1. Does the manuscript provide a valid rationale for the proposed study, with clearly identified and justified research questions?

Reviewer #1: Yes

2. Is the protocol technically sound and planned in a manner that will lead to a meaningful outcome and allow testing the stated hypotheses?

Reviewer #1: Yes

3. Is the methodology feasible and described in sufficient detail to allow the work to be replicable?

Reviewer #1: Yes

4. Have the authors described where all data underlying the findings will be made available when the study is complete?

Reviewer #1: Yes

5. Is the manuscript presented in an intelligible fashion and written in standard English?

Reviewer #1: Yes

6. Review Comments to the Author

You may also provide optional suggestions and comments to authors that they might find helpful in planning their study.

Reviewer #1: Many thanks for your efforts in addressing our feedback or providing rationales in cases where you disagreed with our comments. Overall, we believe the paper is strengthened after this round of reviews. We have no additional feedback. We look forward to reading the full paper with results once available.

The only one minor revision that could be considered is the addition of a word to complete the one sentence (lines 159-162) e.g., '... digital settings"

7. PLOS authors have the option to publish the peer review history of their article (what does this mean? ). If published, this will include your full peer review and any attached files.

**Do you want your identity to be public for this peer review?** For information about this choice, including consent withdrawal, please see our Privacy Policy .

Reviewer #1: No

---

## [Editor Report · Acceptance letter]

PONE-D-24-34705R1

PLOS ONE

Dear Dr. McConnell,

I'm pleased to inform you that your manuscript has been deemed suitable for publication in PLOS ONE. Congratulations! Your manuscript is now being handed over to our production team.

Kind regards,

on behalf of

Prof Catherine Doody

Academic Editor

PLOS ONE